# Boosting Federated Model Convergence with Anomaly Detection and Exclusion

## Abstract

Federated Learning (FL) is becoming increasingly important in AI training, particularly for privacy-sensitive applications. At the same time, it has become a subject of malicious action and needs better protection against adversarial attacks causing data corruptions or other anomalies. In this work, we show that, in contradiction to a popular point of view, if properly introduced security enhancement does improve FL convergence and performance. Taking inspiration from the classical PID control theory, we develop a novel anomaly detection and exclusion approach. Unlike other aggregation techniques that rely solely on current round Euclidean distances between clients, we compute a PID-based history-aware score, which is used to detect anomalies that exceed a statistically defined threshold. Our adaptive exclusion mechanism removes the need for predefined attacker counts, and its server-side linear computational complexity of $O(nd)$ ensures its scalability and practical significance, while existing methods remain superlinear in complexity. We prove theoretically and experimentally verify faster convergence and computational efficiency on several benchmark datasets of various modalities, including non-iid scenarios and different model architectures such as CNNs and LLMs, and show that our method maintains effectiveness while boosting convergence. Our approach is generalizable across diverse task domains and aggregation methods, and is easily implementable in practice.

## 1    Introduction

Federated Learning (FL) is a decentralized Machine Learning (ML) paradigm that enables multiple clients to collaboratively train a shared model without exposing their private data (McMahan et al., 2017). FL is becoming indispensable in modern ML privacy-sensitive applications in various domains, such as healthcare and finance, where data sharing is restricted due to ethical and legal constraints. However, FL systems are prone to learning efficiency degradation due to anomalous client model updates, which can occur because of malicious actions or data corruption (Zhang et al., 2023; Yan et al., 2023). There exist various anomaly detection and exclusion mechanisms operating in the model space (Blanchard et al., 2017b; Mhamdi et al., 2018a; Cao et al., 2021; Shejwalkar et al., 2022), which mitigate the malicious impact but commonly impose computational and communication overhead, potentially slowing down the model's convergence.

We argue that robustness need not come at the expense of efficiency. In fact, we demonstrate that anomaly detection and exclusion can actually enhance FL and lead to models that converge faster and are more accurate. Inspired by the classical Proportional–Integral–Derivative (PID) control theory, we introduce PID-MADE—Proportional-Integral-Derivative Model Anomaly Detection and Exclusion technique. PID-MADE combines three metrics instead of one distance based, employed by competitive methods. While on each round the generic proportional term calculates the current update distance from the target reference, the integral term accumulates the sum of previous distances to capture the persistent misbehavior, and the derivative term - the distance change to flag the sudden attacks. A client is removed only when its composite score breaches a statistically derived threshold, making the method adaptive, history-aware, and lightweight ($O(nd)$ per round). Furthermore, we relax the unrealistic in practice assumptions or system design requirements of some other approaches (Table 1).

Table 1: Comparison of FL aggregation strategies emphasizing convergence and practicality. PID-MADE achieves both **stable/fast convergence** and **practical deployability**.

| Algorithm | Cost | Convergence Speed | Practicality | Advantages (A) and Disadvantages (D) |
|---|---|---|---|---|
| **PID-MADE (ours)** | **Low** $O(nd)$ | **Fast** | **High** | **A:** Scalable, adaptive, stateful, no prior attacker knowledge; ensures convergence stability. **D:** Improper coefficients can make the detection less effective. |
| FedAvg (McMahan et al., 2023) | Low $O(nd)$ | Slow | High | **A:** Extremely cheap. **D:** Easily diverges in adversarial or heterogeneous settings. |
| FedMedian (Yin et al., 2018) | Medium $O(n \log n \cdot d)$ | Medium | High | **A:** Scalable. **D:** Stateless; convergence slower and less consistent. |
| Trimmed Mean (Yin et al., 2018) | Medium $O(n \log n \cdot d)$ | Medium | High | **A:** Scalable. **D:** Requires attacker estimate; stability depends on trimming. |
| FLTrust (Cao et al., 2021) | Low $O(nd)$ | Fast | Low | **A:** Strong in controlled setups. **D:** Impractical without trusted dataset; stateless. |
| RFA (Geometric Median) (Pillutla et al., 2022) | Medium $O(T \cdot nd)$ | Medium | Medium | **A:** Convergence reliable. **D:** Iterative overhead delays training; stateless. |
| Krum (Blanchard et al., 2017a) | High $O(n^2 d)$ | Slow | Low | **A:** Handles many anomalies. **D:** Not scalable; convergence stalls with high variance. |
| Bulyan (Mhamdi et al., 2018b) | High $O(n^2 d)$ | Slow | Low | **A:** Effective filtering. **D:** Extremely costly; convergence often delayed. |
| FoolsGold (Fung et al., 2020) | High $O(n^2 d)$ | Slow | Low | **A:** Handles collusion. **D:** Not scalable; convergence unstable with non-iid data. |
| Clustered FL (Sattler et al., 2020) | Medium $O(nCd)$ | Medium | High | **A:** Improves local convergence. **D:** Not a primary defense; stateless. |
| SignFedAvg (Bernstein et al., 2018) | Low $O(nd)$ | Slow | High | **A:** Very low cost. **D:** Convergence noisy with adversarial or skewed updates. |
| Norm-Clipping (Abadi et al., 2016) | Low $O(nd)$ | Medium | High | **A:** Simple and scalable. **D:** Restricts updates, delaying convergence speed; stateless. |
| FLANDERS (Gabrielli et al., 2024) | Very High $O(max(d^3, m^3))$ | Medium | Low | **A:** Models the temporal evolution of updates using MAR **D:** Cubic complexity challenges wider adoption. |

This paper has the following major contributions. **(1)** We produce a formal theoretical analysis proving that a FL algorithm incorporating model anomaly detection and exclusion (MADE) does not violate the original FL procedure convergence. **(2)** We further provide the theoretical prove and the empirical evidence demonstrating that FL with MADE expedites the convergence in comparison against the undefended FL in the presence of anomalies. **(3)** We introduce PID-MADE, a novel more efficient FL anomaly detection and exclusion technique designed to reduce computational burden while incorporating temporal information about clients. Another key advantage of PID-MADE is that it could be employed with no knowledge of the estimated number of malicious clients or a server-side validation dataset, like in Krum (Blanchard et al., 2017b), FLTrust (Cao et al., 2021) and its derivatives. We prove the PID-MADE's linear computational complexity and provide statistically-grounded recommendations on the threshold selection for FL. **(4)** To facilitate broader adoption and further research, we implement our novel technique as software tools and make them available to the public anonymously[1] for verification. We provide the results of our empirical study and their analysis.

## 2 RELATED WORK

To aggregate the updates, a variety of FL strategies have been developed, ranging from simple but computationally efficient to complex, robust algorithms utilizing more resources to withstand the Byzantine attacks. Table 1 lists the representative sample of the most popular aggregation strategies classified against three characteristics.

**Cost** is primarily evaluated by the server-side computational overhead, where *Low* corresponds to methods with linear complexity (e.g., $O(nd)$), *Medium* is assigned to those with slightly higher superlinear complexity (e.g., $O(n \log n \cdot d)$), and *High* denotes methods with quadratic or higher complexity ($O(n^2 d)$) that are not scalable to the large number of clients.

**Convergence Speed** is the ability of an aggregation method to ensure rapid and stable progress of the global model toward a useful solution from a practical standpoint. *Fast* methods accelerate learning despite potential adversarial interference *Medium* reflects slower or less consistent progress, often requiring more rounds. *Slow* methods converge slowly or stagnate due to either excessive variance introduced by defenses or inability to filter harmful updates effectively.

**Practicality** assesses the feasibility of real-world deployment, where *High* implies a "plug-and-play" nature with no unrealistic assumptions, *Medium* suggests operational hurdles like iterative solvers,

---

[1]https://drive.google.com/file/d/1VSTeE6ynMPQcnGUu_nIZO0_mkQdni8DH/view?usp=drive_link.

| Symbol | Description |
|---|---|
| $i$ | Client index |
| $t$ | Communication round |
| $w_t^i$ | Set of weights sent by client $i$ to the server at round $t$ |
| $\mu_t$ | Model parameters centroid computed by the server at round $t$ |
| $d_t^i$ | Euclidean distance of the $i$-th client model from the centroid at round $t$ |
| $\mathcal{A}$ | Set of all clients participating in the learning process |
| $\mathcal{G} \subset \mathcal{A}$ | Subset of "good" clients not excluded during learning |
| $w_t^{\mathcal{A}}$ | Set of weights for all clients in $\mathcal{A}$ at round $t$: $\mathcal{A} = \{w_t^{i1}, w_t^{i2}, \ldots, w_t^{i|\mathcal{A}|}\}$ |
| $w_t^{\mathcal{G}}$ | Set of weights for all clients in $\mathcal{G}$ at round $t$: $\mathcal{G} = \{w_t^{i1}, w_t^{i2}, \ldots, w_t^{i|\mathcal{G}|}\}$ |
| $w^*$ | Optimal model parameters that minimize the loss function |
| $\mu_t^{\mathcal{G}}$ | Centroid of the "good" models at round $t$ |
| $\mu_t^{\mathcal{A}}$ | Centroid of all models at round $t$ |

Table 2: Notation used in the paper.

and *Low* is reserved for methods with challenging assumptions, such as requiring prior knowledge of the exact number of attackers, or those requiring representative root dataset.

A crucial shared limitation is that all these methods are stateless and rely on unrealistic assumptions. They concentrate on analyzing updates within the scope of a single round, discarding additional information conveyed in the history of client contributions that could reveal long-term or stealth malicious behavior. Additionally, stronger robustness guarantees usually come at a cost of a more complex system overall, like in (Cao et al., 2021; Gabrielli et al., 2024).

# 3 FL Convergence Under Anomalies Formal Analysis

In this section, we present a theoretical analysis to evaluate the feasibility of boosting FL convergence by excluding anomalous clients from the aggregation process. First, we establish that model anomaly detection and exclusion accelerates convergence, and then we quantify the acceleration. Figure 2 summarizes the notation and terminology used throughout the paper. Due to space limitations we present all the proofs in the Appendix.

We propose an FL defense that aims to identify and separate anomalous clients ("bad") from benign ones ("good") to prevent biased model updates that hinder convergence. This is achieved by analyzing the distribution of model updates and removing outliers based on their distance from the server's target reference representation of the optimal model, which is called the *centroid*. This can be described by the following definition of anomalous model weights.

**Definition 1 (Anomalous Model Weights)**: We say that weights submitted by a client are anomalous if they satisfy the following *separation condition*: Assume some training round $t$. The minimal distance between the aggregated anomalous client model updates ($w_t^{\mathcal{A}\setminus\mathcal{G}}$) and the optimal model ($w^*$) must be greater than the maximum distance between the aggregated "good" client model updates ($w_t^{\mathcal{G}}$) and the optimal model, plus the margin $M$: $\min_{\mathcal{A}\setminus\mathcal{G}} \left\| w_t^{\mathcal{A}\setminus\mathcal{G}} - w^* \right\| > \max_{\mathcal{G}} \left\| w_t^{\mathcal{G}} - w^* \right\| + M$, where $M$ is a sensitivity margin for outliers. The defense works as long as $|\mathcal{G}| > |\mathcal{A} \setminus \mathcal{G}|$, i.e. the honest majority persists.

**Criterion 1 (Anomaly Signature in FL)**: In real-world FL deployments, some client updates may deviate drastically from the benign population because of an attacker's poisoned data or simply corrupted measurements. We treat any such persistently "outlying" update as an anomaly. Formally, we say an anomaly in FL satisfies: $\forall \varepsilon > 0, \nexists N \in \mathbb{N}$ s.t. $\forall t \geq N, \left\| w_t^{\mathcal{A}} - w^* \right\| < \varepsilon$. In other words, no matter how small a tolerance $\varepsilon$ we choose, there is no round after which the weights $w_t^{\mathcal{A}}$ remain within that tolerance of $w^*$ due to some anomalous client's contributions. This criterion follows from **Definition 1.**

**Lemma 1 (Variance Reduction through Outlier Removal)**: Let $\{a_i\}$ be a set of points on a number line with scalar values where $a_i \in \mathbb{R}, i \in \mathbb{N}$ and $a_1 < a_2 < \ldots < a_N, N \geq 2$. We consider one of those points, $a_N$, an outlier point $a_o$, meaning that $a_o$ satisfies $(a_o - \mu)^2 = \max_{1 \leq i \leq N}(a_i - \mu)^2$. Let us form a new set of points by simply removing $a_o$ from the original set. Then, if $\sigma^2$ is the variance of the original set and $\sigma'^2$ is the variance of the new set, we have that $\sigma'^2 \leq \sigma^2$.

**Theorem 1 (Convergence Preservation under Anomalous Model Exclusion)**: Consider global models $m^{\mathcal{A}}$ composed by the aggregation of all local models $w_t^{\mathcal{A}}$ and $m^{\mathcal{G}}$ composed by the aggregation of models after exclusion $w_t^{\mathcal{G}}$ through FedAvg. If $\forall \varepsilon > 0, \exists N_1 \in \mathbb{N}$ s.t. $\forall t \geq N_1, \left\| w_t^{\mathcal{A}} - w^* \right\| < \varepsilon$,

then $\exists N_2 \in \mathbb{N}$ s.t. $\forall t \geq N_2, \left\| w_t^{\mathcal{G}} - w^* \right\| < \varepsilon$. That is, assuming the original learning algorithm converges, an algorithm augmented with anomaly detection and exclusion also converges.

**Discussion:** removing anomalous clients from the FL aggregation does not violate the convergence of the original algorithm if it still converges even under the attacks or anomalies. If the original model $m^{\mathcal{A}}$ does converge, this implies that the attack is not strong enough, which in practice can occur due to various reasons, such as a low proportion of malicious clients or the attack goal was to make it converge to the wrong model (Shejwalkar et al., 2022). With the convergence of $m^{\mathcal{A}}$, we can guarantee that if the anomaly detection and exclusion is applied, $m^{\mathcal{G}}$ will always converge to the correct model and faster than $m^{\mathcal{A}}$, which is shown in the next part of the theorem. Furthermore, we make a stronger assumption that even if $m^{\mathcal{A}}$ does not converge, $m^{\mathcal{G}}$ will still converge. While we do not have a theoretical guarantee, this is suggested by empirical evidence, which we present in Sec. 5.

**Theorem 2 (Accelerated Convergence under Anomaly Exclusion):** If $N_1$ is the round, on which the conventional FL with all clients (no clients removed) converges on $w_t^{\mathcal{A}}$, that is $\forall t \geq N_1, \left\| w_t^{\mathcal{A}} - w^* \right\| < \varepsilon$, and $N_2$ is the round, on which FL with good clients only (bad clients are removed) converges on $w_t^{\mathcal{G}}$, that is $\forall t \geq N_2, \left\| w_t^{\mathcal{G}} - w^* \right\| < \varepsilon$, then $N_2 \leq N_1$.

**Discussion:** the implication of this theorem is that, when using only the updates from clients without outlier updates (as in $w_t^{\mathcal{G}}$), the convergence towards the optimal model will be faster than when aggregating updates from all clients, including those with outlier updates (as in $w_t^{\mathcal{A}}$). This is because the outlier updates, which may significantly deviate from the optimal model, distort the global model, causing it to remain far from an optimal solution for a longer period. In Sec. 5, we demonstrate our verification of the convergence on practical use cases. The next result shows a general upper-bound on this accelerated rate (**Theorem 3**).

**Theorem 3 (Enhanced Convergence Rate under Anomaly Exclusion):** The distances between good models' and optimal model weights is bounded by the distances between all models' and optimal model weights, that is $\exists N \in \mathbb{N}$ s.t. $\forall t \geq N, \left\| w_t^{\mathcal{G}} - w^* \right\| \leq C \left\| w_t^{\mathcal{A}} - w^* \right\|$, where $C$ is a constant if the number of malicious clients does not change during learning and $C = \sqrt{\frac{|\mathcal{G}|}{|\mathcal{A}|}} \leq 1$.

*Proof Sketch*: by removing clients sending anomalous updates to the server, we remove the outliers in the weights dimension, which also reduces the variance, as we show in **Lemma 1**. Comparing the variance of benign model weights around the centroid $\mu_t^{\mathcal{G}}$ and the variance of all model weights around $\mu_t^{\mathcal{A}}$, and further rewriting in vector notation using the Euclidean norm, the bound follows by taking the square root.

**Discussion:** the practical significance of this relationship is that the model with weights $w_t^{\mathcal{G}}$ at some round $t \geq N$ will converge quicker than $w_t^{\mathcal{A}}$, and the weights $w_t^{\mathcal{G}}$ will be $\sqrt{\frac{|\mathcal{A}|}{|\mathcal{G}|}}$ times closer to the optimal model than the weights $w_t^{\mathcal{A}}$.

# 4 PID-MADE APPROACH

We develop a novel PID control-inspired algorithm to detect and exclude anomalous updates from FL aggregation. PID provides for a feedback mechanism with three components – proportional, integral, and derivative – widely used in automated control systems since its formalization by (Minorsky, 1922). The goal of the PID controller is to minimize the error value $e(t)$ over time by adjusting the control variable $c(t)$. The error is calculated as the difference between the setpoint and the control variable. The control function $c(t)$ is given by $c(t) = K_p e(t) + K_I \int_0^t e(\varphi) d\varphi + K_d \frac{de(t)}{dt}$, where $e(t)$ is the error value at time $t$, and the coefficients $K_p$, $K_I$, and $K_d$ determine the weights of the proportional, integral, and derivative components.

In our approach, the proportional term reacts to instantaneous deviations, the integral term identifies persistent drifts by accounting for historical trends, and the derivative term anticipates future changes. These components together enable the effective detection of an abnormal client behavior. Our algorithm measures the error as the distance between a client's updates and the *centroid*, which serves as a reference target proxy-optimal model for the server. In the following, we describe how we adopt the PID principle for detecting anomalous clients in FL. The error value is calculated as the

Euclidean distance of client $i$'s model from the centroid $\mu_t$ of all submitted models: $D_t^{(i)}(w_t^{(i)}, \mu_t) = \|w_t^{(i)} - \mu_t\|$. Depending on the robustness requirements of the application, the centroid $\mu_t$ can be either the mean $\frac{1}{N} \sum_{i=0}^{N} w_t^{(i)}$ or the geometric median: $\arg\min_{y} \sum_{i=0}^{N} \|w_t^{(i)} - y\|$. We illustrate our findings in the following sections with the mean variant. The PID score for each client $i$ is calculated as:

$$u_t^{(i)} = \underbrace{K_P D_t^{(i)}(w_t^{(i)}, \mu_t)}_{proportional} + K_I \underbrace{\sum_{x=0}^{t-1} D_x^{(i)}(w_x^{(i)}, \mu_x)}_{integral} + \underbrace{K_d(D_t^{(i)}(w_t^{(i)}, \mu_t) - D_{t-1}^{(i)}(w_{t-1}^{(i)}, \mu_{t-1}))}_{derivative} \cdot$$

For each training round: (1) the server distributes the global model to the clients, (2) the clients train the model locally for a number of epochs and send it back to the server, (3) the server computes the centroid $\mu_t$ and our PID score as detailed above and excludes any clients above the threshold $\tau$, derivation of which we describe in Sec. 4.1. The full mechanism integrated with FedAvg is summarized in Algorithm 1.

---

**Algorithm 1** PID-MADE with FedAVG

---

**Input:** $\mathcal{A}$, set of clients with private local data, alarm rate $\alpha$
 **Output:** $Q$, aggregated global model
**Clients Execute**
  receive global model from the server
  **for** each local epoch **do**
    execute training algorithm (e.g. SGD)
  **end for**
  push the local model $w_t^i$ to the aggregation server
**Server Executes**
  $Q \leftarrow \mathcal{A}$
  **for** each round $t = 1, 2, ...$ **do**
    receive $w_t^i$ from local clients
    compute $\mu_t, u_t^{(i)}, \bar{u}_t, \sigma_t$
    **for** each client $i \in Q$ **do**
      $Q \leftarrow Q \setminus \{w_t^{(i)} : u^{(i)} \leq \tau = \bar{u}(t) + k\sigma_t\}$
    **end for**
    Perform aggregation of weights in $Q$ based on FedAvg.
    Distribute aggregated global model back to the clients.
  **end for**

---

### 4.1 PID-MADE: Anomaly Detection and Exclusion Mechanism

We detect and exclude anomalies by calculating PID scores for each client based on the distance from $\mu_t$ and comparing them against the threshold $\tau$, which is derived from the upper bound of PID scores for non-anomalous clients. Since the optimal model is unknown, we estimate it with the centroid $\mu_t = \frac{1}{N} \sum_{i=0}^{N} w_t^{(i)}$ at each round. This yields a biased estimate in highly non-IID settings, in which case the geometric median can be used instead for more robust estimation (Pillutla et al., 2022). To derive the threshold $\tau$, we analyze the PID metric in Formula 25 and first provide a permissive upper bound which is free from assumptions, but leads to a high false negative rate. To improve this bound, we introduce specific assumptions which allow us to provide a tighter estimate of $\tau$.

**Theorem 4 (Permissive Upper Bound for Benign PID Scores)**: The permissive upper bound of the PID score for the good client is given by $t \cdot \left(\Delta_{max} + O(\frac{f}{N})\right)$, where $\Delta_{max}$ is the maximal deviation from the centroid, $f$ is the number of anomalies, $N$ is the number of all clients, and $t$ is the number of training rounds. This overly permissive bound provides a zero false-positive (benign clients misclassified as malicious) rate, but may yield a high false-negative (malicious clients misclassified as benign) rate. Although impractical as a detection threshold, it serves as a useful baseline from which we derive tighter, more effective bound estimates.

In **Theorems 5** and **6**, we provide tighter and more practical upper bounds on PID scores of non-anomalous clients. Before we introduce **Theorems 5** and **6**, we present **Lemma 2** and **Assumption 1** which are necessary for us to prove the theorems.

**Lemma 2 (Bounded Centroid Shift)**: The centroid shift is bounded by $O\left(\frac{f}{N}\right)$ (see Appendix, proof of **Theorem 4**).

Let $\Delta_t = \|w_t^{(i)} - \mu_t\|$ be the deviation of client $i$'s update from the centroid at any round $t$.

**Assumption 1 (Uncorrelated Deviations)**: The sequence of random variables $\{\Delta_t\}_{t=0}^T$ satisfies $\forall 0 \le t \ne y \le T : \mathrm{Cov}(\Delta_t, \Delta_y) = 0$.

Although **Assumption 1** does not strictly hold in realistic federated settings, nonzero covariances $\mathrm{Cov}(\Delta_t, \Delta_y)$ can only increase the true variance of the PID score – meaning that the threshold we derive will be more conservative. Empirically, we observe that applying the threshold $\tau$ derived under **Assumption 1** sufficiently separates benign from anomalous clients. A fully rigorous threshold would account for each pairwise covariance term, however, estimating all $\mathrm{Cov}(\Delta_t, \Delta_y)$ online would impose significant overhead, and in practice the independence-based approximation already provides a tight, computationally efficient bound, which we derive in **Theorems 5** and **6**.

**Theorem 5 (One-sided Chebyshev Threshold)**: Let us introduce the random variable $U_t$ representing PID scores. Under **Assumption 1** and using **Lemma 2**, without knowing the distribution of PID scores, with probability at most $\alpha$, the PID scores of good clients will be within $z\sigma_t$ of the sample average of PID scores $\bar{u}_t$, where $\sigma_t$ is the standard deviation of PID scores at round $t$. Formally, $\Pr[U_t - \bar{u}_t \ge z\sigma_t] \le \alpha$, where $\alpha = \frac{1}{(1+z^2)}$ represents the desired alarm rate (i.e. false positive rate). With a probability of at least $1 - \alpha$ the benign clients will be under the threshold $\tau = \bar{u}_t + z\sigma_t$. Equivalently, no more than $\alpha$-fraction of benign clients exceed $\tau$. The threshold derivation follows Chebyshev's inequality (see the Appendix).

**Theorem 6 (Gaussian Threshold)**: If we assume $\Delta_t \sim \mathcal{N}(\mu_\Delta, \sigma_\Delta^2)$, then the PID scores become also Gaussian, $U_t \sim \mathcal{N}(\bar{u}_t, \sigma_t^2)$. Then, the exact Gaussian threshold $\tau_{Gauss} = \bar{u}_t + z_{1-\alpha}\sigma_t$ ensures a false positive rate of $\alpha$, where $z_{1-\alpha}$ is the $z$-score corresponding to desired $\alpha$.

**Theorems 5** and **6** give us an opportunity to efficiently select the threshold value based on the detection statistics we want to achieve in practice. To transfer this theoretical foundation into practice and filter out anomalous clients we compute the expected PID score as sample average $\bar{u}_t = \frac{1}{N}\sum_{i=0}^N u_i$. Any $u_t^{(i)}$ greater than $\bar{u}_t + k\sigma_t$ is flagged as an anomaly and excluded from aggregation. As we show in Sec. 5, this empirical threshold estimation is effective even when $\Delta_t$ are not Gaussian, which is often the case in practice. The full algorithm is presented in Algorithm 1, where the input is the set of client models $\mathcal{A}$ and desired alarm rate $\alpha$, and the output is the non-anomalous client set $Q$ and the aggregated model. Unlike previous methods, our PID-based approach is adaptive and does not require prior knowledge of the number of malicious clients. The integral term accumulates historical deviations, making persistent attackers identifiable over time. Additionally, our method has a linear time complexity of $O(nd)$ which we prove in **Lemma 3**.

**Lemma 3 (Computational Complexity)**: Algorithm 1 runs in $O(nd)$ time, where $n$ refers to the number of clients and $d$ is the dimension of the model parameter space. *Proof*: the computation of the centroid $\mu_t$ and of $u_t^{(i)}, \bar{u}_t, \sigma_t$ are linear $O(nd)$, keeping total complexity linear $O(nd)$.

## 5 EMPIRICAL EVALUATION

**Anomaly Model**: Based on **Criterion 1** of model anomalies, introduced in Sec. 3, we implement and evaluate untargeted data poisoning attacks, focusing on a practical case likely in real-world scenarios (Shejwalkar et al., 2022). Importantly, data poisoning serves as a proxy for a broader class of anomalies, capturing not only malicious behaviors but also inadvertent deviations arising from corrupted, mislabeled, or non-representative client data. Thus, our evaluation encompasses both adversarial and non-adversarial sources of model anomalies.

**Datasets:** We evaluate our PID-based approach on four image datasets and one text dataset: our own Intelligent Transportation Systems (ITS), FEMNIST (Caldas et al., 2018), PneumoniaMNIST and BloodMNIST (Yang et al., 2023). In cases of FEMNIST, PneumoniaMNIST and BloodMNIST, the data was divided based on the source distributions from the datasets. For the FEMNIST dataset, the clients studied had 9 classes corresponding to the digits 0-9, each class containing 30-50 images. In the PneumonisMNIST case, we had 8 clients with two imbalanced classes, class 1 consisted of 200-300 samples and class 2 consisted of 600-700 samples. The BloodMNIST dataset contained

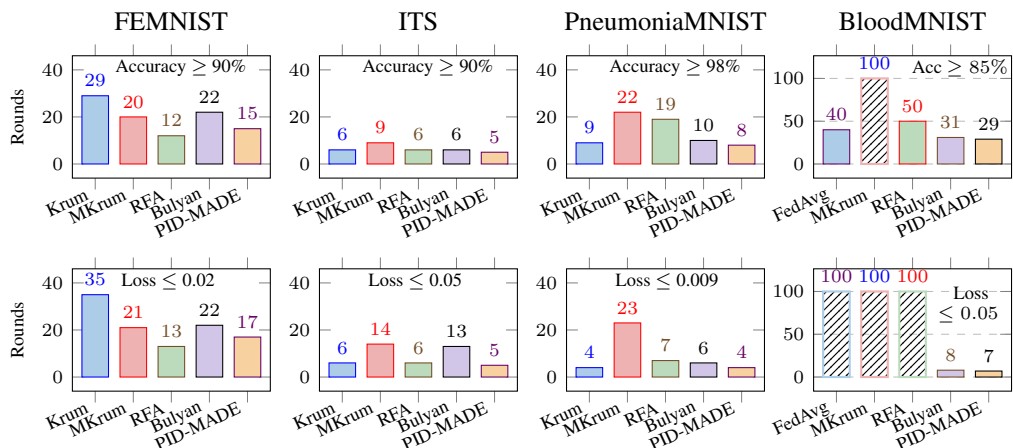

Figure 2: Speed of convergence comparison: the number of rounds required to converge to the specified accuracy level (top) or loss level in our experiments. A dashed bar means the method did not reach the required threshold in the given number of training rounds.

40 clients total. Each client of BloodMNIST had 7 classes with varying samples between 10-70, which illustrated a non-iid case. The ITS datset was designed by us for a binary classification task between "stop sign" and "traffic sign" labels. The images in the "traffic sign" class were purposefully of different size, color, and contained traffic signs in various languages to mimic real-world data that could be collected by an intelligent vehicle. The ITS dataset consisted of 8 clients, with two classes containing 60 samples each. The dataset is fully available with our submission through the code library. For each dataset, in the case of anomalous clients all clients had a poisoning rate of 100%, i.e. all labels were flipped.

The LLM study uses the MedQuAD text dataset (Abacha & Demner-Fushman, 2019), which contains 47,457 medical question-answer pairs curated from 12 NIH websites. The dataset was partitioned into eight FL clients. This partitioning was done in a non-iid manner, simulating realistic heterogeneity in client data distribution across different medical subdomains, with each client LLM learning responses for a specific topic in the medical domain. Some clients also had significantly fewer data points, further contributing to variation in client influence.

**PID-MADE Coefficient Selection:** Figure 1 demonstrates the coefficient selection impact on the model convergence based on our experiments with FEMNIST dataset. Initially, an expert set the coefficients at $K_P = 1, K_I = 0.5$ and $K_d = 0.05$ with expectation of persistent anomalies (yellow line). Additionally, we employed the TPE Bayesian optimization (Bergstra et al., 2011) to tune the coefficients with the goal to expedite the convergence. Other lines on Figure 1 demonstrate how coefficient optimization may speed up the convergence.

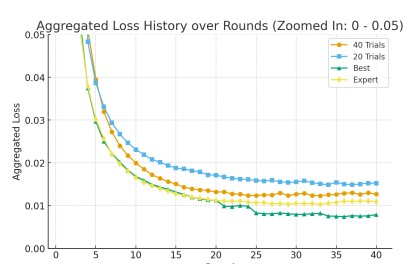

Figure 1: Effect of PID Coefficients selection on the convergence speed.

**Metrics:** We use loss as the metric that illustrates model convergence. To provide a more intuitive comparison of our algorithm against other methods, we define accuracy and loss thresholds. The thresholds help to illustrate how fast each of the algorithms helps achieve convergence practically. These thresholds were selected based on the specific dataset that was studied and the overall performance of the methods in the group, since each dataset presents a different level of challenge.

**Image Classification with CNN Use-Case:** We perform an extensive evaluation of PID-MADE against existing methods on four imaging datasets: FEMNIST, PneumoniaMNIST, our own ITS, and BloodMNIST. This evaluation is performed in ideal, "laboratory" conditions, where we assume that the amount of malicious clients is known a priori, which gives a significant edge to Krum,

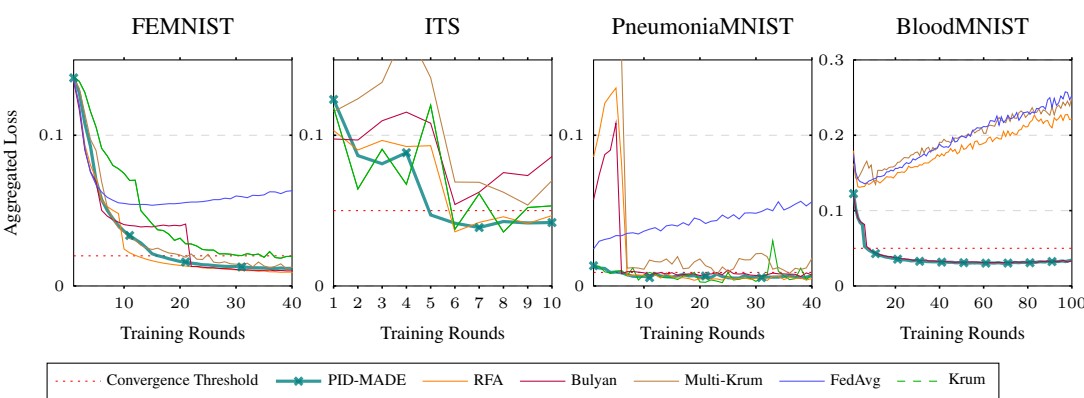

Figure 3: Aggregated loss vs. rounds across datasets. Anomaly detection (PID-MADE) consistently reaches the target loss in fewer rounds.

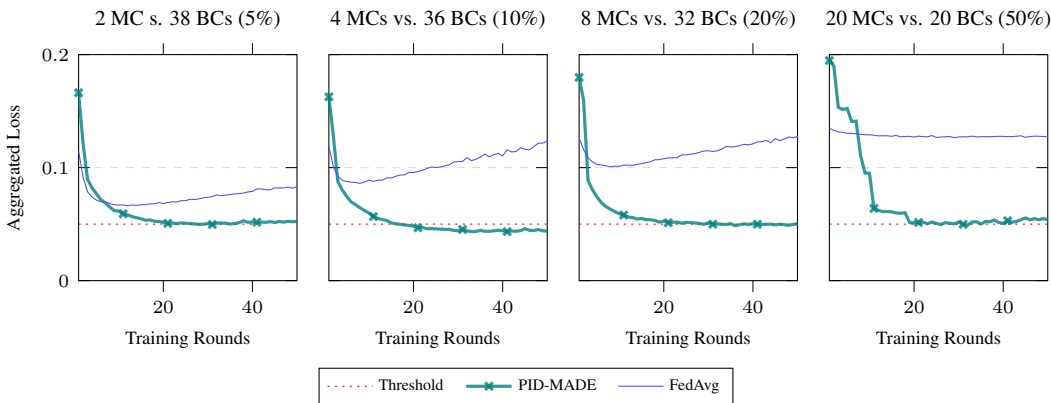

Figure 4: Aggregated loss vs. rounds on Bloodmnist dataset. PID-MADE preserves convergence effectively under varying number of MCs as long as the honest majority persists (three left plots) compared to baseline FedAvg method. With malicious clients reaching 50% convergence becomes problematic

Multi-Krum, and Bulyan. Nonetheless, PID-MADE, without any knowledge about the number of malicious clients, is capable of providing comparable levels of performance. Figure 2 compares the number of communication rounds required by five aggregation methods to reach high accuracy (top row) and low loss (bottom row) on four datasets. Across all experiments, PID-MADE consistently converges faster than the majority of baselines. On ITS, PID-MADE reaches 90% accuracy in only 5 rounds—besting Krum (6), MKrum (9), RFA (6), and Bulyan (6), and achieves the loss threshold in 5 rounds compared with 6–14 for the others. For PneumoniaMNIST, PID-MADE requires just 8 rounds to surpass 98% accuracy (next best is Krum at 9) and ties for the lowest loss convergence (4 rounds, matching Krum). Even on the larger FEMNIST task, PID-MADE cuts the rounds almost in half versus Krum (15 vs. 29 for 90% accuracy; 17 vs. 35 for loss $\leq 0.02$), outperforming MKrum and Bulyan and yielding only to RFA by a small margin. Figure 3 demonstrates the loss curves to achieve the loss thresholds from the bottom row of Figure 2. Together, Figures 2, 3 illustrate that (1) distance-based anomaly detection and exclusion can accelerate convergence and (2) PID-MADE demonstrates faster convergence with no additional knowledge which is required by other approaches (e.g. number of malicious clients).

We further stress-test PID-MADE with increasing the ratio of malicious participants. The results in Figure 4 show that PID-MADE consistently drives the global model below the target loss threshold when benign clients are in majority, whereas FedAvg fails to converge and increasingly diverges as the number of malicious clients grows. While in the borderline case (20 MCs) PID-MADE is less effective, it can drive the model towards convergence, although with more noticeable fluctuations. This highlights PID-MADE's ability to preserve fast and stable convergence even under high adversarial pressure.

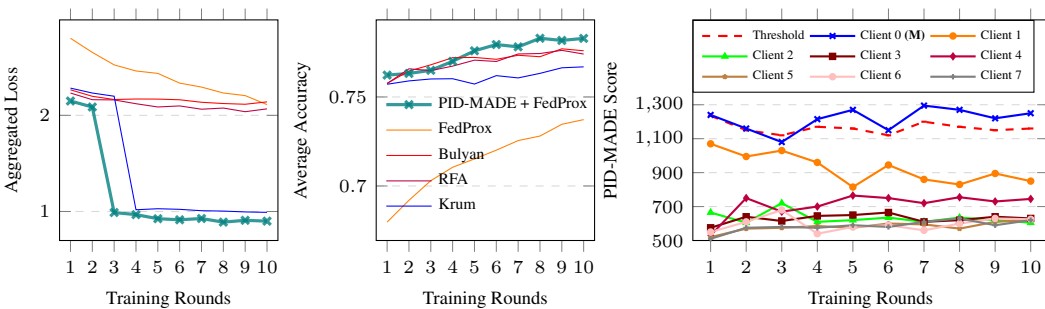

Figure 5: PID-MADE generalization to LLMs. On federated MLM with a domain LLM, MADE-PID maintains stable optimization and preserves accuracy while separating adversarial from benign clients. In (c), Client 0 (**M**) is the only malicious client.

**Masked Language Modeling (MLM) Task with LLM Use-Case:** We illustrate the generalization ability of PID-MADE to the language modeling domain based on the MLM task (Devlin et al., 2019). We utilize FedProx in this case to illustrate how our method can be applied to other aggregation methods besides FedAvg. Figure 5 shows the performance of PID-MADE combined with FedProx (PID-MADE + FedProx) compared to baseline FedProx, Krum, Bulyan and RFA on a federated MLM task with one malicious client injected and non-iid data. On the left, the aggregated loss decreases much more sharply under PID-MADE, whereas the closest Krum still can't achieve the same level of low loss. The center panel highlights the same trend in terms of accuracy: PID-MADE consistently achieves higher average client accuracy throughout training, surpassing 0.75 by round 3 and continuing to improve. Finally, the right panel illustrates the PID-MADE scores per client. Here, the malicious client (Client 0) is cleanly separated from benign participants, with scores remaining above the threshold. This demonstrates that PID-MADE not only accelerates convergence and improves accuracy, but also provides reliable adversarial detection even in the challenging LLM setting with diverse client data in the cross-silo case.

## 6 LIMITATIONS

The statistical nature of our threshold derivation implies that PID-MADE benefits from larger client populations, which may limit its effectiveness in deployments with extremely few clients. Our approach is effective as long as the honest majority of clients persists, which is shown in Figure 4. We assume that benign updates share roughly similar deviation patterns. In highly non-IID environments, where legitimate clients' data distributions vary dramatically, PID-MADE can misclassify rare-but-valid updates as anomalies.

## 7 CONCLUSION

We demonstrated that augmenting FL with anomaly detection and exclusion improves learning efficiency by provably boosting convergence. Our theoretical analysis provided a foundation for understanding how FL anomaly exclusion mechanisms contribute to faster convergence of the global model. As the key contribution, we introduced PID-MADE, a novel FL detection mechanism offering several key advantages over existing approaches. Notably, PID-MADE operates without requiring the estimate of expected anomalies, unlike other methods such as Krum and its derivatives, freeing users from specifying this potentially difficult-to-determine parameter in practice. PID-MADE's theoretical analysis demonstrated linear computational complexity while maintaining similar or even better learning efficiency, a critical factor for scalability in large-scale FL deployments. Finally, we also provided statistically justified recommendations for threshold selection, which were verified empirically and demonstrated improved performance against state-of-the-art methods.

## 8 REPRODUCIBILITY STATEMENT

All code necessary to replicate the experimental results reported in this paper is provided in an anonymized Google Drive repository. The link to the repository is included on the second page of the manuscript and in the supplementary materials to ensure full reproducibility while maintaining

anonymity. The supplementary materials also detail recommended hardware resources to replicate the experiments.

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

## A  PROOFS

### A.1  PROOF OF CRITERION 1

Assume that $\forall \varepsilon > 0, \exists N \in \mathbb{N}$ s.t. $\forall t \geq N$, $\|w_t^{\mathcal{A}} - w^*\| < \varepsilon$. This means that by **Definition 1** $\min_{\mathcal{A} \backslash \mathcal{G}} \|w_t^{\mathcal{A} \backslash \mathcal{G}} - w^*\| < \varepsilon$, which can only happen when there are no anomalies. Hence, we have reached a contradiction with **Definition 1**, and thus for every $\varepsilon > 0$ there is no such $N$ for which $\|w_t^{\mathcal{A}} - w^*\| < \varepsilon$ is satisfied.

### A.2  PROOF OF LEMMA 1

We will show that removing outliers reduces the variance for a set of points on a number line with scalar values. Let $\{a_i\}$ be a set where $a_i \in \mathbb{R}, i \in \mathbb{N}$ and $a_1 < a_2 < \ldots < a_N$. We consider one of those points, $a_N$, an outlier point $a_o$, meaning that $a_o$ significantly deviates from the rest of the points. The mean $\bar{a}$ of $\{a_i\}$ is given as

$$\bar{a} = \frac{1}{N} \sum_{i=1}^{N} a_i. \tag{1}$$

If we exclude $a_o$, the new mean $\bar{a}'$ is

$$\bar{a}' = \frac{1}{N-1} \sum_{i=1}^{N-1} a_i. \tag{2}$$

But (1) can be rewritten as

$$\bar{a} = \frac{1}{N} \left( \sum_{i=1}^{N-1} a_i + a_o \right) \tag{3}$$

$$\bar{a} = \frac{N-1}{N} \bar{a}' + \frac{a_o}{N} \tag{4}$$

Equivalently,

$$\bar{a} - \bar{a}' = \frac{a_o - \bar{a}'}{N} \tag{5}$$

Variance $\sigma^2$ of the set without outlier removal:

$$\sigma^2 = \frac{1}{N} \sum_{i=1}^{N} (a_i - \bar{a})^2 \tag{6}$$

$$\sigma^2 = \frac{1}{N} \left( \sum_{i=1}^{N-1} (a_i - \bar{a})^2 + (a_o - \bar{a})^2 \right) \tag{7}$$

Variance $(\sigma')^2$ of the set with $a_o$ removed:

$$(\sigma')^2 = \frac{1}{N-1} \sum_{i=1}^{N-1} (a_i - \bar{a}')^2 \tag{8}$$

The deviation of each term $a_i$ around the mean $\bar{a}$ is

$$a_i - \bar{a} = a_i - \bar{a}' - (\bar{a} - \bar{a}') \tag{9}$$

Using (5):

$$a_i - \bar{a} = a_i - \bar{a}' - \frac{a_o - \bar{a}'}{N} \tag{10}$$

$$(a_i - \bar{a})^2 = \left( a_i - \bar{a}' - \frac{a_o - \bar{a}'}{N} \right)^2 \tag{11}$$

$$= (a_i - \bar{a}')^2 - 2(a_i - \bar{a}')\left( \frac{a_o - \bar{a}'}{N} \right) + \left( \frac{a_o - \bar{a}'}{N} \right)^2 \tag{12}$$

$$\sum_{i=1}^{N-1} (a_i - \bar{a})^2 =$$
$$\sum_{i=1}^{N-1} (a_i - \bar{a}')^2 - 2\left( \frac{a_o - \bar{a}'}{N} \right) \sum_{i=1}^{N-1} (a_i - \bar{a}') + \tag{13}$$
$$(N-1)\left( \frac{a_o - \bar{a}'}{N} \right)^2$$

$\sum_{i=1}^{N-1}(a_i - \bar{a}') = 0$ due to sum of deviations around the mean being zero. Then (13) reduces to

$$\sum_{i=1}^{N-1} (a_i - \bar{a})^2 =$$
$$\sum_{i=1}^{N-1} (a_i - \bar{a}')^2 + (N-1)\left( \frac{a_o - \bar{a}'}{N} \right)^2 \tag{14}$$

Plugging (14) into (7) we get

$$\sigma^2 = \frac{1}{N} \left[ \sum_{i=1}^{N-1} (a_i - \bar{a}')^2 + (N-1)\left( \frac{a_o - \bar{a}'}{N} \right)^2 + (a_o - \bar{a})^2 \right] \tag{15}$$

Using (8):
$$\sigma^2 = \frac{N-1}{N}\sigma'^2 + \frac{(N-1)}{N}\left( \frac{a_o - \bar{a}'}{N} \right)^2 + \frac{(a_o - \bar{a})^2}{N} \tag{16}$$

Given that $a_o$ is sufficiently large, from (16) it follows that $\sigma^2 > \sigma'^2$.

### A.3 PROOF OF THEOREMS 1 AND 3

According to *lemma 1* (the inequality here is not strict because we might not remove any model weights at all):
$$\frac{1}{|\mathcal{G}|} \sum_{i \in \mathcal{G}} \left( w_t^i - \mu_t^{\mathcal{G}} \right)^2 \le \frac{1}{|\mathcal{A}|} \sum_{j \in \mathcal{A}} \left( w_t^j - \mu_t^{\mathcal{A}} \right)^2 \tag{17}$$

Multiplying by $|\mathcal{G}|$ both sides and additionally multiplying the right side by $\frac{|\mathcal{A}|}{|\mathcal{A}|}$ yields:
$$\sum_{i \in \mathcal{G}} \left( w_t^i - \mu_t^{\mathcal{G}} \right)^2 \le \frac{|\mathcal{G}|}{|\mathcal{A}|} \sum_{j \in \mathcal{A}} \left( w_t^j - \mu_t^{\mathcal{A}} \right)^2 \tag{18}$$

In vector notation using the Euclidean norm:
$$\left\| w_t^{\mathcal{G}} - \mu_t^{\mathcal{G}} \right\|^2 \le \frac{|\mathcal{G}|}{|\mathcal{A}|} \left\| w_t^{\mathcal{A}} - \mu_t^{\mathcal{A}} \right\|^2 \tag{19}$$

Because centroid $\mu_t^{\mathcal{A}}$ minimizes $\left\| w_t^{\mathcal{A}} - \mu_t^{\mathcal{A}} \right\|^2$:

$$\left\| w_t^{\mathcal{G}} - \mu_t^{\mathcal{G}} \right\|^2 \leq \frac{|\mathcal{G}|}{|\mathcal{A}|} \left\| w_t^{\mathcal{A}} - \mu_t^{\mathcal{A}} \right\|^2 \leq \frac{|\mathcal{G}|}{|\mathcal{A}|} \left\| w_t^{\mathcal{A}} - \mu_t^{\mathcal{G}} \right\|^2 \tag{20}$$

$$\lim_{t \to \infty} \left\| \mu_t^{\mathcal{G}} - w^* \right\| = 0$$

For $t \geq N$:

$$\left\| w_t^{\mathcal{G}} - w^* \right\|^2 \leq \frac{|\mathcal{G}|}{|\mathcal{A}|} \left\| w_t^{\mathcal{A}} - w^* \right\|^2 \tag{21}$$

Finally,

$$\left\| w_t^{\mathcal{G}} - w^* \right\| \leq \sqrt{\frac{|\mathcal{G}|}{|\mathcal{A}|}} \left\| w_t^{\mathcal{A}} - w^* \right\| \quad \square \tag{22}$$

### A.4 Proof of Theorem 2

Let assume that possibly $N_2 > N_1 : \exists N_1 < k < N_2$, such that for $w_k^{\mathcal{A}}$ and $w_k^{\mathcal{G}}$, there exists round $k$ such that:

$$\left\| w_k^{\mathcal{A}} - w^* \right\| < \varepsilon \quad \text{and} \quad \left\| w_k^{\mathcal{G}} - w^* \right\| > \varepsilon, \tag{23}$$

meaning that in round $k$

$$\left\| w_k^{\mathcal{A}} - w^* \right\| \leq \left\| w_k^{\mathcal{G}} - w^* \right\| \tag{24}$$

that contradicts to the definition given in (1). In (24), $k$ is sufficiently large such that the outlier updates significantly affect the global model $w_k^{\mathcal{A}}$, causing it to deviate beyond $\varepsilon$-distance from $w^*$.

### Additional commentary to Theorems 1 and 3

If we further split $w_t^{\mathcal{A}}$ into "good" $w_t^{\mathcal{G}}$ and "bad" $w_t^{\mathcal{B}}$ clients ($\mathcal{B} = \{w_t^{i_1}, w_t^{i_2}, \ldots, w_t^{i_{|\mathcal{B}|}}\}$), we can derive the following, more detailed bound for the relation $\frac{\| w_t^{\mathcal{A}} - w^* \|}{\| w_t^{\mathcal{G}} - w^* \|}$:

$$w_t^{\mathcal{A}} = \frac{|\mathcal{G}|}{|\mathcal{B}| + |\mathcal{G}|} w_t^{\mathcal{G}} + \frac{|\mathcal{B}|}{|\mathcal{B}| + |\mathcal{G}|} w_t^{\mathcal{B}}$$

$$\left\| w_t^{\mathcal{A}} - w^* \right\| = \frac{|\mathcal{G}|}{|\mathcal{B}| + |\mathcal{G}|} \left\| w_t^{\mathcal{G}} - w^* \right\| + \frac{|\mathcal{B}|}{|\mathcal{B}| + |\mathcal{G}|} \left\| w_t^{\mathcal{B}} - w^* \right\|$$

Dividing both sides by $\| w_t^{\mathcal{G}} - w^* \|$ yields

$$\frac{\left\| w_t^{\mathcal{A}} - w^* \right\|}{\left\| w_t^{\mathcal{G}} - w^* \right\|} = \frac{|\mathcal{G}|}{|\mathcal{B}| + |\mathcal{G}|} + \frac{|\mathcal{B}|}{|\mathcal{B}| + |\mathcal{G}|} \frac{\left\| w_t^{\mathcal{B}} - w^* \right\|}{\left\| w_t^{\mathcal{G}} - w^* \right\|}$$

In comparison to Theorem 1.2, here we provide an equality, i.e. we can quantify the relation $\frac{\| w_t^{\mathcal{A}} - w^* \|}{\| w_t^{\mathcal{G}} - w^* \|}$. However, since $w^*$ in practice is unknown our approximation can only be based on $\mu_t$. This would further increase the term $\frac{\| w_t^{\mathcal{B}} - w^* \|}{\| w_t^{\mathcal{G}} - w^* \|}$ to $\frac{\| w_t^{\mathcal{B}} - \mu_t \|}{\| w_t^{\mathcal{G}} - \mu_t \|}$.

### A.5 Proof of Theorem 4

Proof: PID score:

$$u_t^{(i)} = D_t^{(i)}(w_t^{(i)}, \mu_t) + K_I \sum_{x=0}^{t-1} D_x^{(i)}(w_x^{(i)}, \mu_x) + K_d (D_t^{(i)}(w_t^{(i)}, \mu_t) - D_{t-1}^{(i)}(w_{t-1}^{(i)}, \mu_{t-1})) \tag{25}$$

where

$$D_t^{(i)}(w_t^{(i)}, \mu_t) = \left\| w_t^{(i)} - \mu_t \right\|.$$

Substitute into (25):

$$u_t^{(i)} = \left\| w_t^{(i)} - \mu_t \right\| + K_I \sum_{x=0}^{t-1} \left\| w_x^{(i)} - \mu_x \right\| + (1 - K_I) \left\{ \left\| w_t^{(i)} - \mu_t \right\| - \left\| w_{t-1}^{(i)} - \mu_{t-1} \right\| \right\} \quad (26)$$

Assume minority of clients are anomalous, i.e. $f < \frac{N}{2}$, where $f$ is the number of anomalies. Let's first show that the centroid $\mu_t = \frac{1}{N} \sum_i w_t^{(i)}$ will not be shifted significantly.

$$\mu_t = \frac{1}{N} \sum_i w_t^{(i)} = \frac{1}{N} \left( \sum_{i \in \mathcal{G}} w_t^{(i)} + \sum_{j \in \mathcal{B}} w_t^{(j)} \right). \quad (27)$$

Consider the purely good centroid $\mu_t^{\mathcal{G}}$:

$$\mu_t^{\mathcal{G}} = \frac{1}{|\mathcal{G}|} \sum_{i \in \mathcal{G}} w_t^{(i)} = \frac{1}{N - f} \sum_{i \in \mathcal{G}} w_t^{(i)}. \quad (28)$$

Let's subtract $\mu_t^{\mathcal{G}}$ from both sides of (4):

$$\mu_t - \mu_t^{\mathcal{G}} = \frac{1}{N} \sum_{i \in \mathcal{G}} w_t^{(i)} - \frac{1}{N - f} \sum_{i \in \mathcal{G}} w_t^{(i)} + \frac{1}{N} \sum_{j \in \mathcal{B}} w_t^{(j)}. \quad (29)$$

$$\mu_t - \mu_t^{\mathcal{G}} = \left( \frac{1}{N} - \frac{1}{N - f} \right) \sum_{i \in \mathcal{G}} w_t^{(i)} + \frac{1}{N} \sum_{j \in \mathcal{B}} w_t^{(j)}.$$

Take the Euclidean norm on both sides and apply triangle inequality:

$$\left\| \mu_t - \mu_t^{\mathcal{G}} \right\| \leq \left| \frac{1}{N} - \frac{1}{N - f} \right| \sum_{i \in \mathcal{G}} \left\| w_t^{(i)} \right\| + \frac{1}{N} \sum_{j \in \mathcal{B}} \left\| w_t^{(j)} \right\|. \quad (30)$$

Note that

$$\left| \frac{1}{N} - \frac{1}{N - f} \right| = \left| \frac{N - f - N}{N(N - f)} \right| = \frac{f}{N(N - f)}. \quad (31)$$

Now, **assume** both anomalous and benign norms are bound with some constant $\zeta$. This means that $\sum_{i \in \mathcal{G}} \left\| w_t^{(i)} \right\| \leq (N - f)\zeta$ and $\sum_{j \in \mathcal{B}} \left\| w_t^{(i)} \right\| \leq f\zeta$. Considering this and (31), we rewrite (30) as:

$$\left\| \mu_t - \mu_t^{\mathcal{G}} \right\| \leq \frac{f(N - f)}{N(N - f)} \zeta + \frac{f}{N} \zeta \leq \frac{2f}{N} \zeta \leq O\left( \frac{f}{N} \right). \quad (32)$$

(32) Provides an upper bound on the centroid shift, which refer to as **Bounded Centroid Shift** in **Lemma 2** of the main paper. This lemma allows us to bound the PID score for benign clients. Let's analyze each term in (8) by rewriting it with the benign centroid $\mu_t^{\mathcal{G}}$:

$$u_t^{(i) \in \mathcal{G}} = \underbrace{\left\| w_t^{(i) \in \mathcal{G}} - \mu_t \right\|}_{Proportional} + K_I \underbrace{\sum_{x=0}^{t-1} \left\| w_x^{(i) \in \mathcal{G}} - \mu_x \right\|}_{Integral} + (1 - K_I) \underbrace{\left\{ \left\| w_t^{(i) \in \mathcal{G}} - \mu_t \right\| - \left\| w_{t-1}^{(i) \in \mathcal{G}} - \mu_{t-1} \right\| \right\}}_{Derivative}.$$

$$(33)$$

First, consider the proportional part. Add and subtract $\mu_t^{\mathcal{G}}$:

$$\left\| w_t^{(i) \in \mathcal{G}} - \mu_t + \mu_t^{\mathcal{G}} - \mu_t^{\mathcal{G}} \right\| \leq \left\| w_t^{(i) \in \mathcal{G}} - \mu_t^{\mathcal{G}} \right\| + \left\| \mu_t - \mu_t^{\mathcal{G}} \right\| \leq \left\| w_t^{(i) \in \mathcal{G}} - \mu_t^{\mathcal{G}} \right\| + O\left( \frac{f}{N} \right). \quad (34)$$

Assuming bounded heterogeneity between good clients, the upper bound on $\left\| w_t^{(i) \in \mathcal{G}} - \mu_t^{\mathcal{G}} \right\|$ is some $\Delta_{max}$. Then (34) is bounded above by $\Delta_{max} + O(\frac{f}{N})$.

Second, look at the integral part. Same manipulation:

$$\sum_{x=0}^{t-1} \left\| w_x^{(i) \in \mathcal{G}} - \mu_x + \mu_t^{\mathcal{G}} - \mu_t^{\mathcal{G}} \right\| \leq \sum_{x=0}^{t-1} \left\| w_x^{(i) \in \mathcal{G}} - \mu_t^{\mathcal{G}} \right\| + \sum_{x=0}^{t-1} \left\| \mu_x - \mu_t^{\mathcal{G}} \right\| = \sum_{x=0}^{t-1} \left\| w_x^{(i) \in \mathcal{G}} - \mu_t^{\mathcal{G}} \right\| + t \cdot O\left(\frac{f}{N}\right) = t\left(\Delta_{max} + O\left(\frac{f}{N}\right)\right)$$

(35)

Third, we do the same analysis on the derivative part:

$$\left\| w_t^{(i) \in \mathcal{G}} - \mu_t \right\| - \left\| w_{t-1}^{(i) \in \mathcal{G}} - \mu_{t-1} \right\| \leq \left\| w_t^{(i) \in \mathcal{G}} - \mu_t^{\mathcal{G}} \right\| + O\left(\frac{f}{N}\right) - \left\| w_{t-1}^{(i) \in \mathcal{G}} - \mu_{t-1}^{\mathcal{G}} \right\| + O\left(\frac{f}{N}\right) \leq O\left(\frac{f}{N}\right).$$

(36)

Combining (34), (35), (36), we get that for a good client, the upper bound on PID value $u_t^{(i) \in \mathcal{G}}$ is $\Delta_{max} + O\left(\frac{f}{N}\right) + t\left(\Delta_{max} + O\left(\frac{f}{N}\right)\right) + O\left(\frac{f}{N}\right)$, which can be simplified to $t\left(\Delta_{max} + O\left(\frac{f}{N}\right)\right)$.

The permissive upper bound of a threshold for the PID score of good clients $\{i : i \in \mathcal{G}\}$ is $t\left(\Delta_{max} + O\left(\frac{f}{N}\right)\right)$. This upper bound ensures zero false positive rate, however, the false negative rate can be expected to be high. This bound is not usable, however, it provides a starting point for us to derive a more tight and practical threshold.

### A.6 PROOF OF THEOREM 5

*Proof:*

$$\mathbb{E}[U_t] = \mathbb{E}\left[\|w_t^{(i)} - \mu_t\|\right] + K_I \sum_{x=0}^{t-1} \mathbb{E}\left[\|w_x^{(i)} - \mu_x\|\right] + K_D \mathbb{E}\left[\|w_t^{(i)} - \mu_t\| - \|w_{t-1}^{(i)} - \mu_{t-1}\|\right]$$ (37)

Using $\mu_\Delta = \mathbb{E}[\Delta_t]$

$$\mathbb{E}[U_t] = \mu_\Delta + K_I t \mu_\Delta + 2K_D \mu_\Delta \approx \mu_\Delta(1 + K_I t)$$ (38)

Next, derive the variance of our PID Score. Due to Bienaymé identity we would have additional covariance terms, but those can be neglected due to assumption (1). The variance of $U_t$ then becomes:

$$\sigma_t^2 = \text{Var}[U_t] = \text{Var}[\Delta_t] + K_I^2 \sum_{x=0}^{t-1} \text{Var}[\Delta_x] + 2K_D \text{Var}[\Delta_t] = \sigma_\Delta^2 + K_I^2 \sigma_\Delta^2 + 2K_D^2 \sigma_\Delta^2$$ (39)

Finally, using Chebyshev's inequality we can state that with a probability of at least $1 - \alpha$ the benign clients will be under the threshold $\tau = \bar{u}_t + z\sigma_t$. Equivalently, no more than $\alpha$-fraction of benign clients exceed $\tau$.

### A.7 PROOF OF THEOREM 6

*Proof:* the Gaussian threshold follows directly from **Theorem 5** under the Gaussian assumption $\Delta_t \sim \mathcal{N}(\mu_\Delta, \sigma_\Delta^2)$. If $\alpha$ is the desired alarm rate, than using the standard normal distribution and the $z$-score corresponding to $1 - \alpha$ gives us $\Pr[U_t > \tau_{Gauss}] = 1 - \Phi(z_{1-\alpha}) = \alpha$.

# B  PERFORMANCE

## B.1  PID COMPUTATION TIME

Table 3: Time required for computing the PID score and RFA criterion as the number of participating clients increases from 5 to 20 clients

| Total clients | MADE-PID Time (ms) | RFA (next fastest, ms) |
|---|---|---|
| 5 | 2.225 | 88.404 |
| 6 | 2.394 | 75.425 |
| 7 | 2.160 | 75.405 |
| 8 | 2.575 | 75.462 |
| 9 | 2.602 | 93.799 |
| 10 | 2.757 | 83.496 |
| 11 | 3.051 | 95.252 |
| 12 | 3.348 | 99.076 |
| 13 | 3.771 | 106.084 |
| 14 | 3.752 | 106.784 |
| 15 | 4.126 | 116.48 |
| 16 | 4.260 | 112.684 |
| 17 | 4.196 | 118.06 |
| 18 | 4.329 | 132.287 |
| 19 | 4.559 | 130.349 |
| 20 | 4.767 | 142.596 |

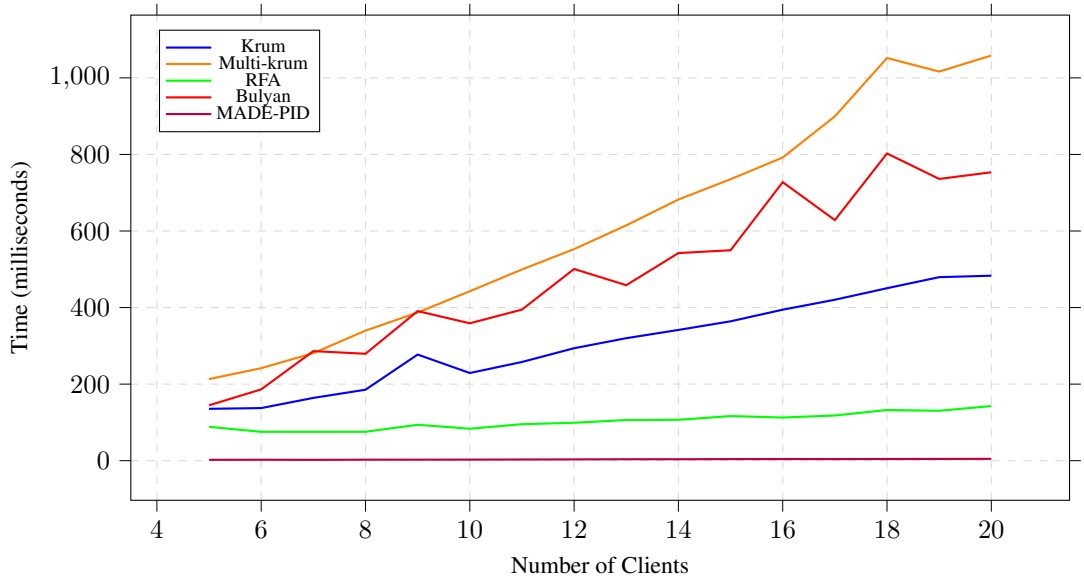

Figure 6: Metric computation time based on the number of clients

Figure 7: Computational complexity study - metric calculation time growth as the number of total clients increases

# C   ADDITIONAL RESULTS

Table 4: Attack Mitigation Effect with PID-MADE vs. FedAvg baseline. FEMNIST (2/20).

| Poisoning rate (%) | Total FP | Total FN | PID-MADE Accuracy (%) | FedAvg Accuracy (%) |
|---|---|---|---|---|
| 10% | 0 | 3 | 96.8 | 93.22 |
| 50% | 0 | 1 | 95.7 | 92.83 |
| 100% | 0 | 0 | 96.8 | 89.90 |

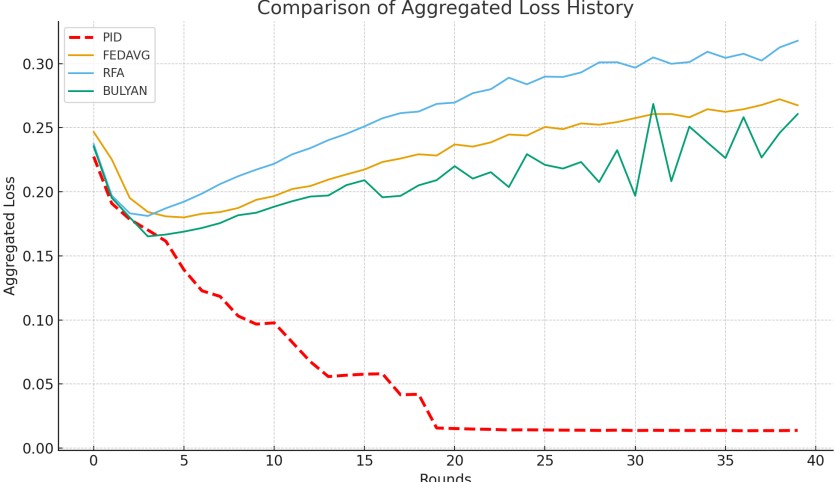

Figure 8: 100 clients with full participation. 10 clients were poisoned with full label flipping.

# D   RECOMMENDATIONS FOR EXPERT COEFFICIENT SELECTION

In cases where automated coefficient selection is not possible, we provide the following recommendations. The proportional coefficient should be fixed at $K_P = 1$, since this term reflects information from the current round—similar to what is used in existing methods such as Multi-Krum and RFA.

The integral coefficient $K_I$ should be increased (e.g., $K_I \in [0.5, 1]$) in scenarios where detecting slow, low intensity consistent attacks is critical. The derivative coefficient $K_D$ should also fall within $[0.5, 1]$ only when detecting sudden anomalous updates becomes important, otherwise it can introduce additional fluctuations into the overall PID score. We provide the following edge case classification which can help guide PID coefficient selection.

- **Edge case (1)**, $K_P = 1, K < 0.5, K_D < 0.5$: majority of the clients are malicious, the attack is implemented by randomly shifting (through model or data poisoning) each $w_t^{(i)}$ that is controlled by the attacker. Depending on the selection of $\tau$, our PID-MADE algorithm will still work, but may yield false positives and false negatives.

- **Edge case (2)** $K_P = 1, K \in [0.5, 1], K_D < 0.5$: minority of the clients are malicious, but they perform a coordinated attack. In this case the centroid that serves as a proxy for $w^*$ should be calculated as the geometric median. Since the geometric median is known to be statistically robust to outliers, PID-MADE is still effective and can identify malicious clients.

- **Edge case (3)**: If malicious clients constitute a majority and execute a coordinated attack that leads to targeted shifts of the centroid position (as well as in edge case (2)), our technique, like other similarity-based detection methods (e.g., cosine similarity, Euclidean distance), will not be effective.

# E    CLIENT DATA DISTRIBUTIONS

These plots illustrate client data distributions

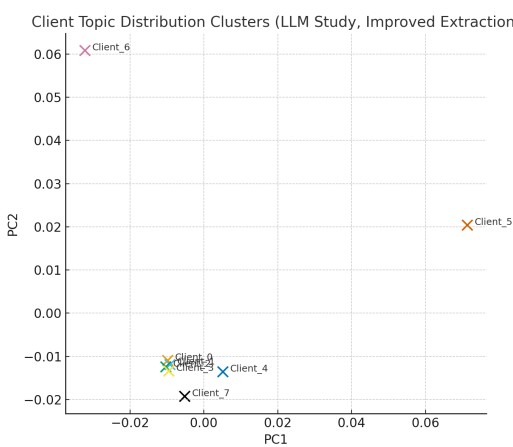

Figure 9: LLM study: each client was assigned a different topic, but some topics shared similarities.

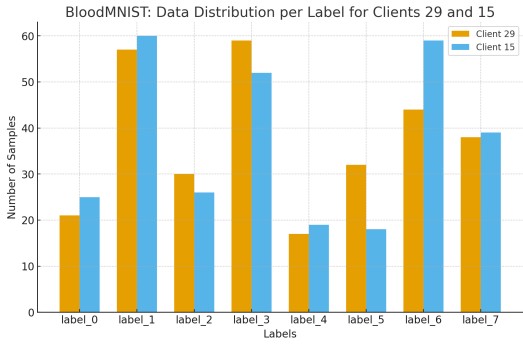

Figure 10: Example of moderate non-iid clients in BloodMNIST dataset.

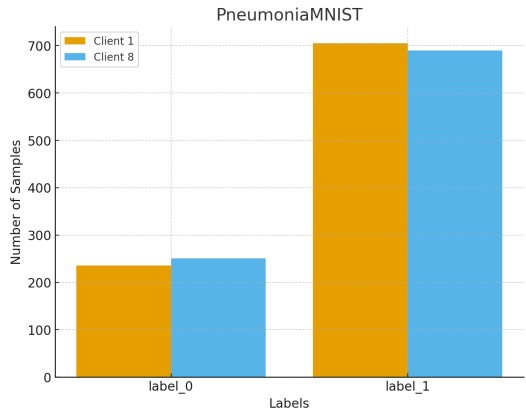

Figure 11: IID label imbalance in PneumoniaMNIST dataset

# F CODE AND DATASET ARTIFACTS

The experiments were conducted on a system equipped with an AMD Ryzen 5 7600 CPU, 32 GB of RAM, and an NVIDIA RTX 4060TI GPU with 16 GB of dedicated memory, running the Ubuntu 22.04 OS. Our code may be used for the reproduction and further reconfiguration of our experimental setup. Additionally, it provides the ability to collect and save metrics necessary for the further analysis. We also provide the datasets that we used to facilitate the reproduction of our empirical study experiments. All the shared materials can be found by this anonymized link that does not disclose the authors' identities: https://drive.google.com/file/d/1VSTeE6ynMPQcnGUu_nIZO0_mkQdni8DH/view?usp=drive_link.

Datasets used in the experiments are initially downloaded from AWS by the execution script and later can be found in the *datasets/* folder of the archive.

Guidelines for the experiment setup configuration and execution are included in the README with the code artifacts found in the link above.

## G    LLM USAGE

An LLM was used to assist with enhancing the communication in the paper. All research results and contributions presented in this paper are original human work.

