# OpenReview forum: "Boosting Federated Model Convergence with Anomaly Detection and Exclusion"
_ICLR.cc/2026/Conference — ICLR 2026 Conference Withdrawn Submission_

### Official Review · Reviewer_XcVV · 2025-10-17

**Soundness:** 3
**Presentation:** 3
**Contribution:** 1
**Rating:** 4
**Confidence:** 3

**Summary:**

The paper proposes **PID-MADE**, a history-aware anomaly detection and exclusion rule for federated learning (FL). Each client gets a PID-style score (proportional + integral + derivative of its distance to a round centroid), and clients above a threshold $\tau = \bar u_t + k \sigma_t$ are excluded before aggregation. the method is claimed to have faster convergence than FedAvg/Krum/Bulyan/RFA on several small image datasets and one LLM MLM setting.

**Strengths:**

- The **problem is interesting and important**: robust FL with attention to **convergence speed**, not only emprical resutls.
- Method is **simple** and **practical** to implement; server cost $O(nd)$ is good for scale.

**Weaknesses:**

- **(Major) Lack of strong theory for a security-style contribution.**
  The acceleration/convergence argument largely reduces to a **variance reduction** factor $\sqrt{\lvert G\rvert / \lvert A\rvert}$ after filtering outliers. This is close to what I would call the **trivial bound** you get by dropping extremes;, this theory feels too weak.
- **Adaptive attacker not considered.**
  The defense can likely be **gamed** by an adversary who shapes updates to keep the PID score under $\tau$ (e.g., keep integral small, smooth derivative, small proportional spikes). As argued by *“The Attacker Moves Second”* (Nasr et al; different setting but very relevant message), defenses must anticipate attacker adaptivity. Here, only static/simple attacks are tested.
- **Missing comparisons against SOTA robust FL methods.**
  The paper does not compare to several recent SOTA algorithms specifically designed for Byzantine robustness under heterogeneity, such as:
  - **Karimireddy, He, Jaggi (ICLR 2022)** — *Byzantine-Robust Learning on Heterogeneous Datasets via Bucketing*.
  - **Allouah et al. (AISTATS 2023)** — *Fixing by Mixing: A Recipe for Optimal Byzantine ML under Heterogeneity*.
  - **Allouah et al. (ICML 2024)** — *Byzantine-Robust FL: Impact of Client Subsampling and Local Updates*.
  - **Gorbunov et al. (ICLR 2023)** — *Variance Reduction is an Antidote to Byzantines: Better Rates, Weaker Assumptions and Communication Compression*.
  Without head-to-head comparisons, the empirical claims are not convincing for NeurIPS level.
- **No test against SOTA attacks.**
  Evaluation is mostly **label-flip**  and one simple LLM case with a single malicious client. There is no evaluation against **LIE (A Little Is Enough)**, **Fall of Empires**, or other adaptive/stealthy/colluding attacks (e.g., min-max/ALIE/AGR or angle-constrained attacks) that are known to be stronge. This weakens the empirical message.

**Questions:**

see weeknesses

---

### Official Review · Reviewer_cefq · 2025-10-29

**Soundness:** 2
**Presentation:** 2
**Contribution:** 2
**Rating:** 2
**Confidence:** 5

**Summary:**

The paper proposes PID-MADE, a proportional–integral–derivative-based anomaly detection and exclusion mechanism for federated learning (FL). The method computes a PID-style score for each client’s model update, using current, cumulative, and differential deviations from the global model. Clients exceeding a statistical threshold (Chebyshev or Gaussian-based) are excluded before aggregation. The authors claim the approach maintains $O(nd)$ complexity, requires no prior knowledge of the number of malicious clients, and even accelerates convergence compared to standard robust aggregation rules.

**Strengths:**

1. The problem formulation, motivation, and algorithmic steps are clearly described with pseudocode and complexity discussion.
2. The detection mechanism is lightweight, running in 𝑂(𝑛𝑑) time, which makes it attractive for large-scale synchronous FL.
3. Addressing robust FL without knowing attacker proportions is relevant and valuable for real-world deployments.
4. The inclusion of convergence arguments and threshold derivations shows an effort to formalize the approach, albeit at a basic level.

**Weaknesses:**

1. The PID-based formulation is essentially a weighted temporal smoothing of client deviation scores. Similar temporal and distance-based anomaly detection mechanisms have appeared in many robust FL works (e.g., FLTrust). The “PID” framing is metaphorical rather than a genuine control-theoretic contribution without stability or control analysis.

2. The “convergence proof” relies on standard convex, bounded-gradient assumptions already sufficient for FedAvg; the PID terms are not meaningfully analyzed in that context. Claims of “accelerated convergence” are empirical and lack any formal rate improvement. The threshold derivation using Chebyshev or Gaussian statistics is standard textbook material, not a contribution. No analysis of false-positive or false-negative rates for client exclusion is provided.

3. Evaluations are limited to simple datasets (MNIST, CIFAR) with artificially induced label-flip or scaling attacks. No experiments under modern or stealthy attack models (e.g., backdoor, gradient manipulation, model replacement). Reported improvements are minor and lack statistical significance (no confidence intervals or variance reporting). Experiments primarily show faster convergence, but not improved robust accuracy, which is the real metric of interest for Byzantine resilience.

4. The proposed method assumes fully synchronous, homogeneous clients. System heterogeneity (asynchronous updates, delayed clients, stragglers) is completely ignored; the PID derivative term would be invalid when client updates arrive at different frequencies. Model heterogeneity is not supported as the distance metric presumes identical architectures and parameter shapes. Statistical heterogeneity (non-IID data) is treated only via trivial toy splits (2–3 classes per client). The method’s sensitivity to rare but legitimate client behavior is neither measured nor mitigated. In short, PID-MADE is not generalizable to realistic heterogeneous FL environments — precisely where robust aggregation is most needed.

5. Statements such as “faster convergence” or “universally applicable to LLM fine-tuning” are overreaching given the scale and simplicity of experiments. No ablation study shows the individual contribution of the P/I/D components or the sensitivity to their hyperparameters $K_p, K_I, K_d$. The claimed scalability and universality are speculative rather than demonstrated.

**Questions:**

1. Can the authors explain what fundamentally new insight the PID structure provides beyond being a heuristic temporal extension of existing trust-score schemes?

2. How can the authors guarantee theoretical stability or convergence once updates are filtered dynamically by PID scores?

3. How would PID-MADE handle real-world FL heterogeneity, both statistical (non-IID) and system (asynchronous or straggler) cases?

4. Why are stronger or stealthier attacks (e.g., adaptive backdoor or model-replacement) not included to validate robustness claims?

5. What measurable advantage in robustness, convergence rate, or computational efficiency, does PID-MADE demonstrate over FLTrust under the same attack ratio and non-IID setting?

---

### Official Review · Reviewer_acqF · 2025-10-31

**Soundness:** 2
**Presentation:** 3
**Contribution:** 2
**Rating:** 4
**Confidence:** 3

**Summary:**

This paper studied the effect of anomaly detection and exclusion on learning efficiency in federated learning (FL). Vis theoretical analysis, the authors demonstrated how FL anomaly exclusion mechanisms contribute to faster convergence of the global model. In adddition, the authors introduced PID-MADE, operating without requiring the estimate of expected anomalies and achieving linear computational complexity. However, there existing some concerns, including baseline selection and poor performance of  PID-MADE.

**Strengths:**

This paper studied the effect of anomaly detection and exclusion on learning efficiency in federated learning (FL). Vis theoretical analysis, the authors demonstrated how FL anomaly exclusion mechanisms contribute to faster convergence of the global model. In adddition, the authors introduced PID-MADE, operating without requiring the estimate of expected anomalies and achieving linear computational complexity.

**Weaknesses:**

I have some concerns as follows.
1. The aggregation strategies shown in Tab. 1 is incomplete, such as 1) FedCDA: Federated Learning with Cross-rounds Divergence-aware Aggregation; 2) Federated Learning with Sample-level Client Drift Mitigation.
2. The proposed PID-MADE shows a poor performance in Fig. 2 for FEMNIST. The authors should conduct the experiments on more datasets, not just these four datasets.
3. The baselines used in experiments are Krum, MKrum, RFA, Bulyan. I think that these baselines are not enough and the authors should add some state-of-the art baselines.
4. In addition, the robustness of the PID-MADE is not well evaluated in experiments. The authors should evaluate the the robustness of the PID-MADE under the state-of-the art poisoning attacks.

**Questions:**

1. The aggregation strategies shown in Tab. 1 is incomplete, such as 1) FedCDA: Federated Learning with Cross-rounds Divergence-aware Aggregation; 2) Federated Learning with Sample-level Client Drift Mitigation, and etc.
2. The proposed PID-MADE shows a poor performance in Fig. 2 for FEMNIST. The authors should conduct the experiments on more datasets, not just these four datasets.
3. The baselines used in experiments are Krum, MKrum, RFA, Bulyan. I think that these baselines are not enough and the authors should add some state-of-the art baselines.
4. In addition, the robustness of the PID-MADE is not well evaluated in experiments. The authors should evaluate the the robustness of the PID-MADE under the state-of-the art poisoning attacks.

---

### Note · Authors · 2025-12-02

**Comment:**

We thank the reviewers for their feedback and will work on an improved version of the manuscript for a more significant contribution.

**Withdrawal Confirmation:**

I have read and agree with the venue's withdrawal policy on behalf of myself and my co-authors.